# Telomere analysis using 3D fluorescence microscopy suggests mammalian telomere clustering in hTERT-immortalized Hs68 fibroblasts

Nancy Adam[1], Erin Degelman[2], Sophie Briggs[1], Rima-Marie Wazen[3], Pina Colarusso[3,4], Karl Riabowol[1,5]* & Tara Beattie[1]*

Telomere length and dynamics are central to understanding cell aging, genomic instability and cancer. Currently, there are limited guidelines for analyzing telomeric features in 3D using different cellular models. Image processing for telomere analysis is of increasing interest in many fields, however a lack of standardization can make comparisons and reproducibility an issue. Here we provide a user's guide for quantitative immunofluorescence microscopy of telomeres in interphase cells that covers image acquisition, processing and analysis. Strategies for determining telomere size and number are identified using normal human diploid Hs68 fibroblasts. We demonstrate how to accurately determine telomere number, length, volume, and degree of clustering using quantitative immunofluorescence. Using this workflow, we make the unexpected observation that hTERT-immortalized Hs68 cells with longer telomeres have fewer resolvable telomeres in interphase. Rigorous quantification indicates that this is due to telomeric clustering, leading to systematic underestimation of telomere number and overestimation of telomere size.

[1] Department of Biochemistry & Molecular Biology, University of Calgary, Calgary, AB, Canada. [2] Cumming School of Medicine, University of Calgary, Calgary, AB, Canada. [3] Live Cell Imaging Laboratory, Cumming School of Medicine, University of Calgary, Calgary, AB, Canada. [4] Department of Physiology & Pharmacology, University of Calgary, Calgary, AB, Canada. [5] Department of Oncology, University of Calgary, Calgary, AB, Canada. *email: karl@ucalgary.ca; tbeattie@ucalgary.ca

Telomeres are specialized forms of chromatin at the ends of linear chromosomes, consisting of the repetitive (TTAGGG)$_n$ sequence in vertebrates[1]. In humans, these repetitive sequences are bound predominantly by six proteins termed the shelterin complex, composed of TRF1, TRF2, Pot1, TPP1, Rap1, and TIN2. The resultant specialized nucleoprotein structure plays an important role in preventing chromosomes from being recognized as one-sided double strand breaks (DSBs)[2]. The shelterin complex shields the physical telomere end through facilitating formation of a telomere (T)-loop, in which the single-stranded 3′end overhang folds back into the duplex array[3], displacing one strand to form a displacement (D)-loop. It is thought that when telomere sequences shorten to a critical length, a DNA damage response is triggered which leads to activation of ATM[4], p53[5] and downstream molecules such as p21 to block further cell replication. This results in a permanent cell cycle arrest called replicative senescence. Senescence can then be thought of as a first line of defense against cancer since it blocks cells from becoming genomically unstable.

Human telomeres lose approximately 50–100 bp/cell division of their telomeric sequences due to the end replication problem[6–8]. The loss of telomeric DNA can result either in cellular senescence seen in normal cells or in genomic instability in cancer cells in which senescence is circumvented and cells continue to divide[9]. Therefore, average telomere length has been used as a surrogate to measure the replicative capabilities of cells and is proposed to be a reliable biomarker of aging[10–12]. However, studies have shown that average telomere length may not be an accurate read out for replicative senescence and that a subset of short telomeres may be responsible for signalling senescence, telomere dysfunction and cellular fate[13–15]. Furthermore, there is heterogeneity in telomere length among individuals, among cell types of the same individual and even among different cells of the same tissue, which raises questions of whether average telomere length, telomere length heterogeneity, or telomere integrity are most important in triggering these cellular processes[16].

Several techniques to measure absolute or relative telomere lengths have been developed[17]. One standard method to measure the telomere length of individual chromosomes is quantitative-fluorescence in situ hybridization (qFISH)[18]. In this procedure, a peptide nucleic acid (PNA) probe conjugated to a fluorophore is used to specifically label telomeric DNA. The probe generates a fluorescence signal that is proportional in intensity to the length of the telomere and can be used to estimate the relative lengths within the same cell. qFISH is often used to examine telomeres in metaphase spreads, which allows for the staining of individual chromosomes and their identification if they are labelled with chromosome-specific probes. Detailed observations of telomere intensities using this technique revealed that the telomeres of subsets of chromosomes can be quite short in some strains of normal cells and that telomeres begin to fuse upon depletion of members of the shelterin complex.

While studying telomeres in two-dimensional (2D) metaphase spreads is a powerful approach, it is important to localize and characterize telomeres in three-dimensional (3D) in interphase cells given that interphase cells constitute the great majority of most somatic cell types. Using conventional optical microscopy techniques such as widefield and confocal microscopy, several studies have provided fundamental insights into the 3D organization of telomeres in each cell cycle phase and how this is altered in cancer cells[19,20]. Telomeres appear to have a spherical shape, they can form aggregates and have a volume of approximately 0.01 μm$^3$ that varies with the cell type and telomere length[21,22]. More recently, super-resolution microscopy approaches using PNA probes conjugated to Alexa-647 fluorophores have been able to visualize the relatively small T-loop structure on chromatin spreads[23]. Using similar single molecule localization microscopy techniques, the measured telomere size in interphase cells was reported to have a radius of approximately 60–400 nm translating to a volume of 0.002–0.01 μm$^3$ depending on cell type and telomere length[24–27].

Although super-resolution technologies are powerful for visualizing detailed structures, they are not yet readily available to all researchers, are costly and labour intensive. Furthermore, super-resolution offers less throughput especially when imaging multiple colours. While the size of telomeres is close to the diffraction limit, the smallest telomere can still be visualized using wide-field microscopy if the fluorescence signal is bright enough to be detected. Therefore, wide-field microscopy may be a screening method before moving onto the more complex imaging techniques. Furthermore, wide-field microscopy lends itself to multi-colour imaging in 3D and fast acquisition, which is important for investigating telomere spatial organization within nuclei, including the relation to other components such as the shelterin complex and specific chromatin domains.

To standardize telomere analysis in interphase cells, we have developed a user-friendly guide for studying telomeres in 3D using wide-field microscopy. We describe technical details in a pedagogical manner so that researchers with limited microscopy experience can quickly and efficiently record and analyse data sets. We illustrate the important steps in image acquisition and analysis by investigating normal diploid fibroblast cells, which have traditionally been used in the study of telomere biology using other methods[8], yet have not been examined using this 3D approach before. Fibroblasts are mesenchymal cells that play important roles in wound healing, synthesizing extracellular matrix and producing tissue stroma. In humans, each cell contains 23 pairs of chromosomes leading to a total of 92 telomeres/diploid cell. Since telomere numbers are constant, relative measurements can be performed on telomeres in interphase fibroblast cells. This paper describes a reproducible, rigorous and accessible workflow that is practical to many cell types, analysis software packages and microscopy systems for telomere analysis. To demonstrate the application of the methods discussed, two biological samples containing known numbers of telomeres with different lengths, Hs68 fibroblasts and Hs68+ human telomerase reverse transcriptase (Hs68-hTERT), are investigated.

## Results

**Image acquisition.** Hs68 fibroblasts were used to develop a standardized workflow for investigating telomeric features and samples were prepared according to the FISH protocol described in the methods section. Wide-field microscopy was used for image acquisition and a flow chart summarizing the procedures described below is shown in Fig. 1 (Supplementary Fig. 1 for detailed protocol). Wide-field microscopy is sometimes considered inferior to laser scanning confocal microscopy regarding resolution, yet camera-based systems are more sensitive, cost-effective, and can be more time efficient when imaging in 3D[28]. Furthermore, wide-field microscopy, when combined with current powerful digital deconvolution methods, yields optical sections comparable to the optical sectioning inherent in confocal techniques. Another advantage is that the camera technologies, such as scientific complementary metal–oxide–semiconductor (sCMOS) cameras, allow researchers to acquire larger fields-of-view, thus reducing experimental time at the wide-field microscope[29].

To ensure consistency and reproducibility, samples were imaged using the same microscope settings, including illumination intensity, exposure time and step size of Z-stacks. When acquiring images, it is important to recognize the intensity values

## I. FISH staining

Telomeric PNA probe
DAPI staining

## II. Image Acquisition

Immunofluorescence Wide-field Microscope
Z-stack of sample

## III. Pre-processing

a. Darknoise and flatfield correction
b. Huygens Deconvolution

## IV. Image Analysis

a. ImageJ 2D
- Count
b. Imaris 3D
- Intensity
- Volume
- Position

**Fig. 1** Flow chart summarizing procedures described in this study. For a detailed and stepwise protocol, see Supplementary Fig. 1.

that correspond to the number of emitted photons detected by photosites on the camera. The intensity values are expressed over a range that is defined by the bit-depth ($N$) of the camera to describe the amount of grayscale that can be represented in the final image ($2^N$). For example, a 12-bit camera will read out pixel intensity values ranging from 0 to 4095 ($=2^{12}$). Standard practice for immunofluorescence fixed imaging is to fill 50–75% of the bit depth of the camera. The pixel intensity distribution can be gauged by the intensity histogram in the imaging software and is dependent on the exposure time and intensity settings. Pixel saturation was avoided since it leads to misinterpretation of intensities as it does not reflect the actual number of photons

hitting the photosites in the camera. When recording volumetric data, the step size of Z-stacks should capture all fluorescence signal in a 3D space and this can be achieved by imaging at the Nyquist rate in both lateral (X, Y) and axial (Z) directions[30]. Furthermore, to ensure the entire sample is imaged, the top and bottom limits of the Z-stacks should be set to record dim and out-of-focus fluorescence. If these conditions are set incorrectly, subsequent image pre-processing steps, such as deconvolution, will be prone to artefacts[31]. During our imaging, a step size of 70 nm was used for X, Y and 100 nm for Z.

To accurately count telomeres in interphase, PNA staining of a metaphase spread derived from the cells of interest was performed[18]. This is an important step to ensure that the PNA probe binds specifically at the chromosome ends under the performed experimental conditions and to confirm the number of telomeres per cell. This metaphase assay will also reveal if any chromosomes appear to have telomere-free ends or end-to-end fusions[32], which would reduce the count of predicted telomeres in interphase cells. Furthermore, both sister telomeres in a metaphase spread should have the same staining intensity since the majority of sister telomere pairs have equal lengths[33]. In our experiments, when performing a metaphase spread of Hs68 fibroblasts, 46 chromosomes were stained, and each chromatid-end was telomere positive (Fig. 2a, b). This reconfirms that our normal diploid fibroblast cells contain 92 telomeres while in the G0 and G1 phases of the cell cycle. There was no additional background staining of the PNA probe measured since the telomere hybridizations were specific as shown by metaphase spread hybridizations (Fig. 2).

**Image pre-processing**. To ensure that the fluorescence emission intensity is proportional to the number of PNA molecules binding to telomeric repeats, the raw images acquired by the microscopy system must be corrected for non-uniformities arising from uneven illumination, optical artefacts, and inhomogeneous sensor response as well as the noise that artificially increases the intensity values measured by the camera. This procedure is known as background correction and is crucial when comparing pixel intensities of different samples.

A background correction requires that additional microscope-specific images are recorded, which include a dark frame image and a flat-field image. In practice, raw images of Hs68 cells in each channel were acquired first which contain the uncorrected pixel fluorescence intensity values (Fig. 3a). Next, a dark frame was acquired by using the same exposure time as the raw image with the camera shutter closed (Fig. 3b). The dark frame pixel intensities are usually low, and it is standard to average the intensity values of 9–16 dark frames to get a mean pixel value that can be used for the correction. With our imaging set-up, the average pixel value in a dark image was on the order of six counts. Next, a flat-field image was captured (Fig. 3c) to correct for the uneven illumination resulting from across the array of the detector. A flat-field image is obtained by imaging a uniform fluorescent sample such as concentrated dye solution or 1-mm-thick fluorescent plastic slide[34,35]. When a line was drawn across the flat-field image from the upper left corner to the bottom right corner (Fig. 3d), the profile demonstrates that the illumination used in our system was inhomogeneous across the field of view (Fig. 3d). This is not unexpected because most wide-field systems are configured for illumination that follows a Gaussian distribution; a circular distribution with the highest intensity in the centre that falls towards the edges of the field of view. This fall off near the edges is more pronounced in the latest version of cameras such as sCMOS that allow the capture of larger field of views.

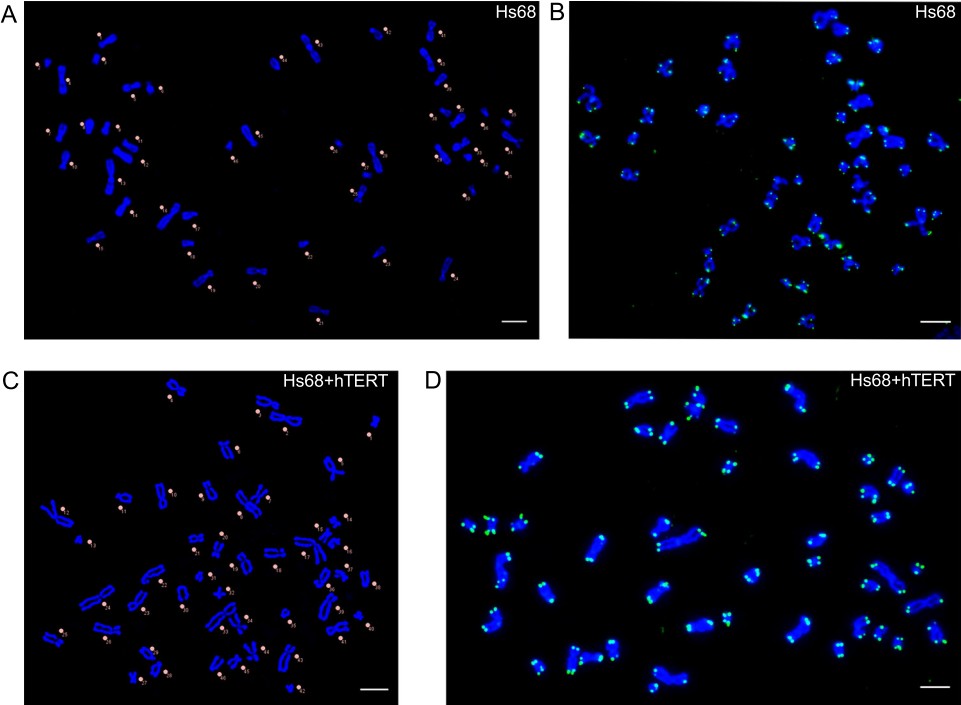

**Fig. 2** Metaphase spreads of **a**, **b** Hs68 and **c**, **d** Hs68 + hTERT cells. DNA was stained with DAPI (blue) and telomeres were stained with PNA-TelC488-conjugated probe (green). **a**, **c** Chromosomes were manually counted and labelled in ImageJ. Both Hs68 and Hs68 + hTERT cells counted 46 chromosomes confirming a diploid genome. **b**, **d** Telomere staining indicated no telomere-free ends in both Hs68 and Hs68 + hTERT cells. Images were acquired using a Nikon Ti Eclipse Widefield system and deconvolved. Scale bar = 5 μm.

The corrected image is generated by performing image calculations according to the following expression:

$$\text{Corrected image} = M\frac{(\text{raw} - \text{dark})}{(\text{flat} - \text{dark})},$$

where $M$ is a scaling constant, and raw, dark and flat are the original uncorrected image, averaged dark frame image and the flat-field image, respectively.

Although the image correction can be performed with many different software packages such as MATLAB or Python, we used ImageJ/FIJI[36] which is an open source image processing package and is accessible for a life science researcher with little programming experience. To preserve numeric accuracy when dividing the raw image by the flat-field image, 32-bit floating was used since the division can result in low pixel intensity values (e.g. between 0.01 and 1 rather than the typical 0 and 4095 for a 12-bit camera). In order to restore the corrected images, a scaling factor was used to stretch the pixel intensity values so that they fill the bit depth of the camera. With our imaging set-up, a constant factor of 3000 was chosen for $M$ so that the resulting pixel intensity values were scaled more reasonably to match the bit depth of our 12-bit camera. The final created images were corrected for optical defects and noise present in the raw images which allows for accurate and reproducible quantitative image analysis of fluorescence intensity (Fig. 3e).

**Counting telomere foci in ImageJ (2D).** To analyse features of telomeres in an interphase cell in ImageJ, a Z-stack that captured the entire nucleus was used. First, a maximum intensity projection was applied to visualize the acquired Z-stack as a 2D-image. The telomeres appeared as bright features against a dark background and an automatic segmentation was used to consistently and conveniently count the number of telomeres per interphase

cell (Fig. 4a). We used the 'Find Maxima' command in ImageJ to identify the local maxima of intensity in an image. The noise tolerance option was adjusted to a value where approximately one point was recognized per telomere. Since Hs68 cells have a diploid genome, we adjusted the noise tolerance to achieve on average 92 telomeres per cell in G1 phase. Cells in S and G2, measured by their DNA content, were excluded from the analysis. With our wide-field images, a noise tolerance of 10 resulted in an average of 93 telomeres per Hs68 cell (Fig. 4b). To ensure the identified maxima represent real telomeres, researchers should acquire images with a high signal-to-noise-ratio (SNR) and confirm the telomere count with their metaphase spread. Although the maximum intensity projection is a convenient representation of the data sets, it is important to recognize that projection of a 3D stack may overlay two features that are separated in the axial (Z) direction. When applied to interphase cells, this would mean that the number of telomeres could be under-estimated.

To measure telomere area and intensity in ImageJ, it is important to segment the foreground, the telomeres, from the background. In fluorescence microscopy, a common image segmentation approach is based on pixel intensity wherefore the out-of-focus fluorescence inherent to wide-field microscopy must be minimized. Deconvolution was used to re-assign out-of-focus fluorescence and it is important to verify whether a particular deconvolution technique preserves the intensity relationships among the different pixels in the image[31]. For the telomere intensity measurements in this study, the corrected images were followed by iterative deconvolution using Huygens Essential Software (Scientific Volume Imaging version 18.10, see Methods). Full iterative deconvolution on wide-filed images significantly improves both contrast and resolution. Applying intensity-based thresholding, Otsu in this case[37], to images before deconvolution can lead to overestimation of the size and merging

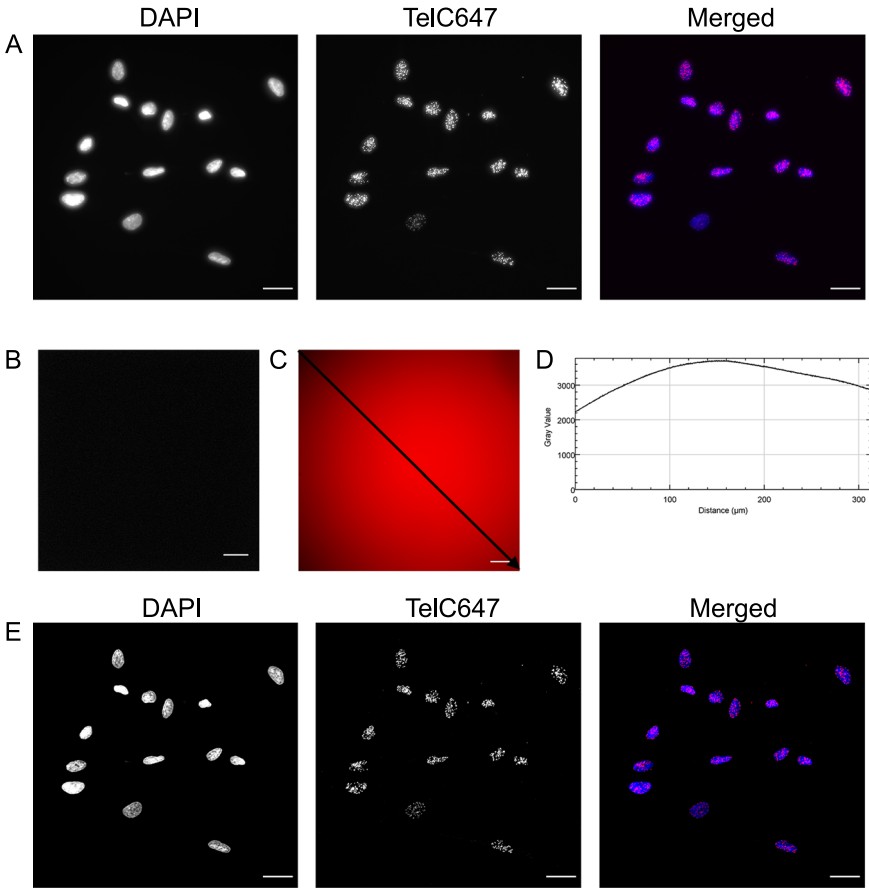

**Fig. 3** Correction of raw images using dark frame and flat-field images. **a** Raw images were acquired using the full field of view (FOV) of the camera. Hs68 cells were stained with DAPI (blue) and PNA-TelC647-conjugated probe (magenta). **b** Averaged dark frame image. **c** Flat-field image showing variable intensity across the field of view. **d** Plot profile intensity graph of the line drawn across the flat-field image in panel **c**. **e** Deconvolved images of the dark noise and flat-field corrected images of panel **a**. All images were taken using a Nikon Ti Eclipse Widefield system. Scale bar = 25 µm.

telomeres together making them appear larger (Fig. 4c). After deconvolution, an improved segmentation is obtained using the same Otsu threshold (Fig. 4c). However, even after deconvolution and watershed segmentation[38], some telomeres are not completely separated and are not recognized as individual spots (Fig. 4d, e). The resolution limit of wide-field systems may lead to two telomeres appearing as one. Drawing a line through points and using the 'Plot Profile' setting in ImageJ will display a graph of intensities of pixels along the line and any differences in gray values would identify individual telomeres (Fig. 4e, f). This feature is also a measure of the spatial lateral (X,Y) resolution of your system, as it reveals the minimum distance for two spots to be successfully resolved as individual spots. If the telomeres are within ~200 nm of each other, they can no longer be resolved as individual spots.

**Analysing telomeres in 3D**. Telomeres in interphase cells are 3D structures, so visualization and analysis of the acquired images in 3D will provide additional biologically meaningful information. Here we describe a detailed approach that gives insight into common settings, concepts and parameters that should be carefully considered when performing 3D analysis. There are many software packages available that are suitable to analyse telomeres in 3D such as ImageJ/FIJI plug-ins TANGO[39] or SpotDistance[40], MATLAB, Metamorph[41] or other 3D licensed programmes such as TeloView[TM19]. Even though there are small differences between these programmes, we recommend applying the same approach and workflow to ensure

reproducibility across researchers and labs. In our lab, the image analysis software Imaris (Bitplane, version 9.2.1) was available and was therefore used to measure telomere length and size in 3D.

Since telomeres are different in size due to the variability in length, it is essential to use intensity as a threshold to grow the spot size accordingly. Similar to ImageJ, this intensity value is used to separate the foreground, the telomeres, from the background. The lowest intensity value of the pixels that represent telomeric signal in an image (Fig. 5a) was determined in Imaris and used as a threshold to accurately represent the telomere in X, Y and Z without overestimating its size (a value of 800 in our case). The 'Spot' function in Imaris can be used to automatically model and detect point-like structures in data and visualize them as spheres. The most accurate approach to locate telomeres is by calculating a mathematical point using a Gaussian filter. A Gaussian filter operates on the intensity distribution to determine a centroid of a given extended object in the image[42]. Within Imaris, the 'Quality' setting represents this mathematical approach and is used to determine the centre of the telomeres (Fig. 5b).

When analysing telomeres in 3D, it is important to calibrate the system using beads to determine the theoretical and measurable smallest volume to ensure accurate volume measurements of telomeres in the images. For our analysis, we imaged 100 nm TetraSpeck™ beads (Life Technologies) and determined axial calibration[43], as well as the calculated volume by the used analysis software (Supplementary Fig. 2, Supplementary Table 1).

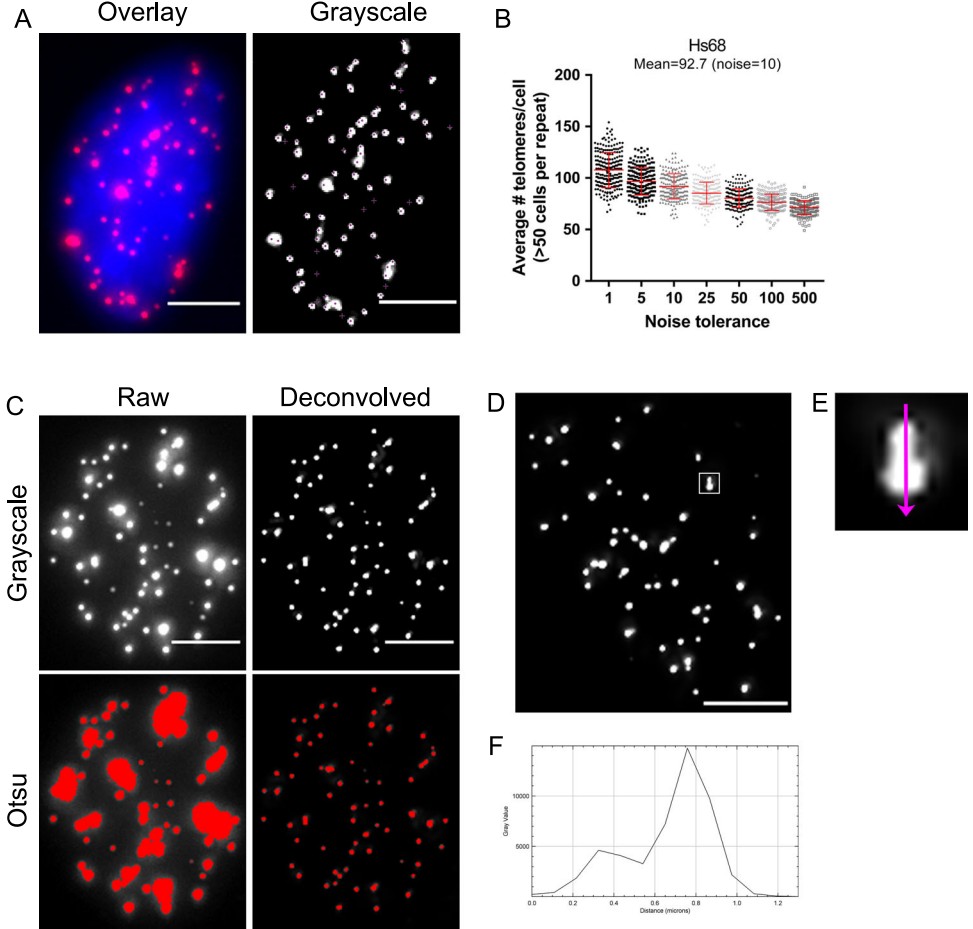

**Fig. 4** 2D analysis of telomeres in interphase Hs68 cells using the ImageJ program. **a** Example of an interphase Hs68 cell stained with PNA-TelC647-conjugated probe for telomeres (magenta) and DAPI (blue). Grayscale, deconvolved image of PNA TelC647 staining was used for telomere count using the Find Maxima program in ImageJ. **b** Graph showing average telomere counts per cell using different noise tolerances for Find Maxima (n = 3 biologically independent experiments, >50 cells per experiment). **c** Grayscale, raw image of PNA TelC647 staining and grayscale, deconvolved image of PNA TelC647 staining. Intensity-based thresholding (Otsu) of raw image and deconvolved image was performed. Red indicates the foreground signal that is separated from the background when applying the Otsu threshold. **d** Grayscale, deconvolved image of PNA TelC647 staining in an Hs68 cell. **e** Inset is an enlargement of the boxed area in panel **d**. **f** Plot profile along the purple arrow in the inset in panel **e**. The intensity graph indicates two spots, but they cannot be separated by thresholding and watershedding. All images were taken using a Nikon Ti Eclipse Widefield system. Scale bars = 5 µm. Source data used for the graphs in **b** can be found in Supplementary Data 1.

It is important to acknowledge the point spread function of a diffraction-limited system and the elongation in Z-axis. This will impact the volume measurements of 3D objects, as the calculated volume will only represent the intensity signals in the images rather than the actual volume of the biological structures. The theoretical lateral resolution is on average three times greater than the theoretical axial resolution as determined by Abbe's formulas, which is dependent on the wavelength and NA[44]. Within Imaris, the values for estimated X, Y and Z are user-set and this lateral-to-axial ratio is then used to determine the spot volume. We used the theoretical lateral (220 nm) and axial (660 nm) values, giving a 3× elongation in the Z-axis. Therefore, all the final spots assigned will have an elliptical shape (Fig. 5c) and the volume measurements can only be used as a relative measurement to compare samples. Even though this set ratio is a major limitation of the software, when the same settings are applied to each sample, the telomeres in the samples can still be compared. After completion of the Spot Wizard function in Imaris, spots are assigned to the telomeric signals (Fig. 5d). Several telomeric features can now be investigated, such as telomere localization within the nucleus, telomere size, telomere intensity and telomere count.

**Preparation of Hs68 cells with elongated telomeres**. To demonstrate the utility of this workflow, we compared young Hs68 fibroblasts (passage 40) with a telomerase-expressing Hs68 strain with elongated telomeres (passage 40 after infection). Hs68 cells expressing telomerase were generated by infecting young Hs68 cells (passage 25) with a pBABE$_{puro}$-FLAG hTERT retrovirus construct to stably express telomerase and induce telomerase-mediated telomere elongation. The expression of hTERT was measured by western blot and quantitative PCR (qPCR) (Fig. 6a, b, Supplementary Fig. 3). While Hs68 fibroblasts contain low levels of hTERT, which has been demonstrated before[45], upon hTERT overexpression both RNA and protein levels were significantly increased. Furthermore, Telomerase Repeated Amplification Protocol (TRAP) confirmed active telomerase activity only in Hs68 + hTERT (Fig. 6c). To assess the biological effects of actively expressed hTERT, telomere length was measured by telomere qPCR (telo-qPCR) and Telomere Restriction Fragments (TRF) analysis. Hs68 + hTERT cells that had been passaged 40 times after transfecting with hTERT expression construct were calculated to have 3.5 times more telomeric sequences than the parental Hs68 cells (Fig. 6d, e). A metaphase spread was performed to confirm a diploid genome

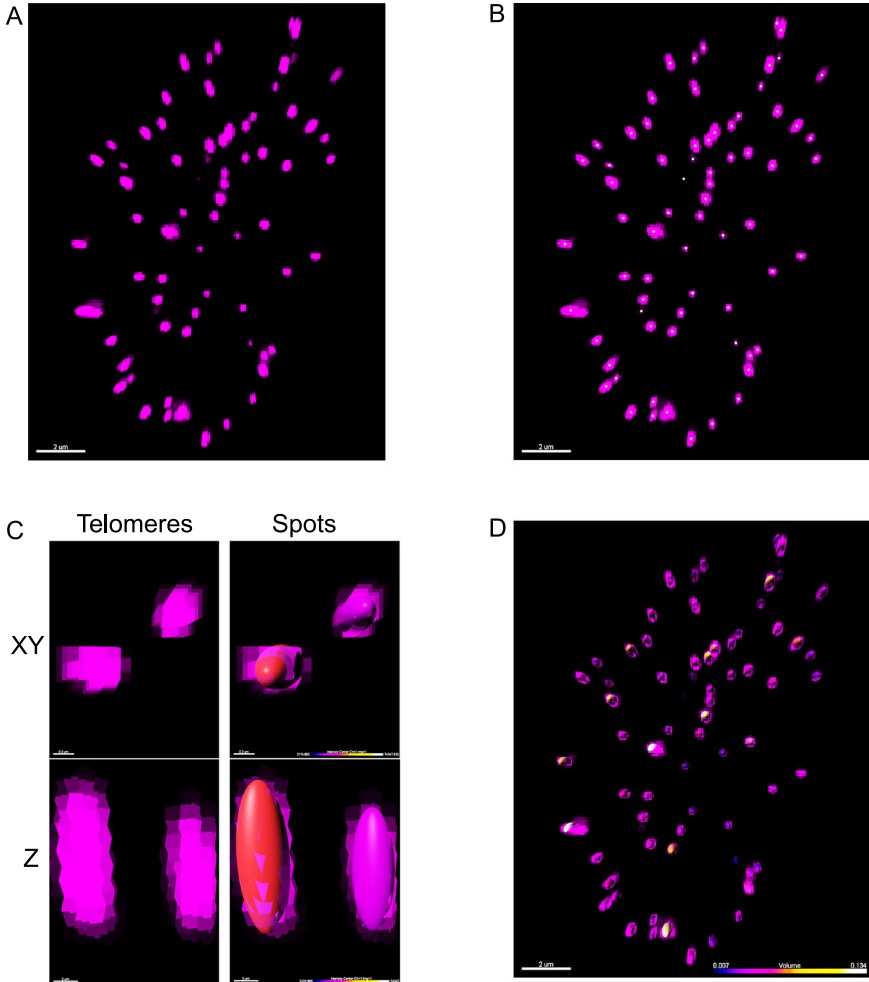

**Fig. 5** Analysing telomeres in 3D using Imaris. **a** Example of a deconvolved, 3D image of an Hs68 cell stained with PNA TelC647-conjugated probe (magenta) visualized in the Imaris program. **b** The centre of the fluorescence signal (small dots) was detected based on Gaussian filtering in Imaris. This was used to determine the location of each telomere within the image. **c** Visualization of the fluorescence signal of two telomeres (magenta) from above (XY) and from the side (Z). Spots were assigned to the fluorescence intensity that represents telomeric PNA probe. Overlay of the spots with the fluorescence intensity from above (XY) and from the side (Z) shows how accurate the assigned spots reflect telomeric signal. **d** Overlay of the spots with the fluorescence intensity representing the telomeres in this 3D image of an Hs68 cell. Spots were colour-coded based on volume and visualized from above. Scale bars = 2 μm.

and no telomere-free ends were seen in Hs68 + hTERT cells (Fig. 2c, d).

**Fluorescence intensity accurately measures telomere length**. The sum of the fluorescence intensities can be used as a readout for telomere length, since the sum represents the total amount of PNA probes bound along the telomere. This technique has been used to study telomere length in large scale studies[46–48]. When analysing telomere length based on intensity sum in Imaris in 3D, Hs68 + hTERT cells were calculated to have longer telomeres when compared to Hs68 (Fig. 7a, b, Supplementary Table 1). When dividing the intensity sum of Hs68 + hTERT by Hs68, the ratio was calculated to be 3.95 ± 0.54 which closely matches data obtained by qPCR and TRF. This indicates that the intensity of the spots in 3D is an accurate readout for relative telomere lengths.

**Telomere volume is not directly proportional to length**. Next, using the intensity sum to interpret the telomere length of individual telomeres, we investigated whether the volume of telomeres is correlated to their predicted lengths. First, we

measured and plotted the volume of all telomeres measured in Hs68 and Hs68 + hTERT. Hs68 cells have telomeres that are on average 0.05 μm$^3$ in volume whereas Hs68 + hTERT cells have telomeres that are on average 0.1 μm$^3$, a twofold increase (Fig. 7c, Supplementary Table 1). Assuming telomere folding is not influenced by telomere length, we would expect the volume of telomeres to change proportionally with the length. However, when plotting the volume against the length of individual telomeres in both Hs68 and Hs68 + hTERT, we did not observe a linear correlation (Fig. 7d, e). Instead, only a weak coefficient of determination $R^2$ was observed of 0.7 and 0.6 for Hs68 and Hs68 + hTERT, respectively. Furthermore, telomeres of a certain length (indicated by Intensity Sum on the X-axis) can appear as different sized foci. This indicates that telomeres are dynamic structures and can adopt different sizes, which potentially depends on additional factors that affect their compaction and tertiary structure. When comparing the coefficient of determination of Hs68 + hTERT ($R^2 = 0.6$) to the $R^2$ value of Hs68 ($R^2 = 0.7$), even a weaker correlation was found. This indicates that longer telomeres may have different folding and/or densities than shorter telomeres, leading to more varied compaction and size.

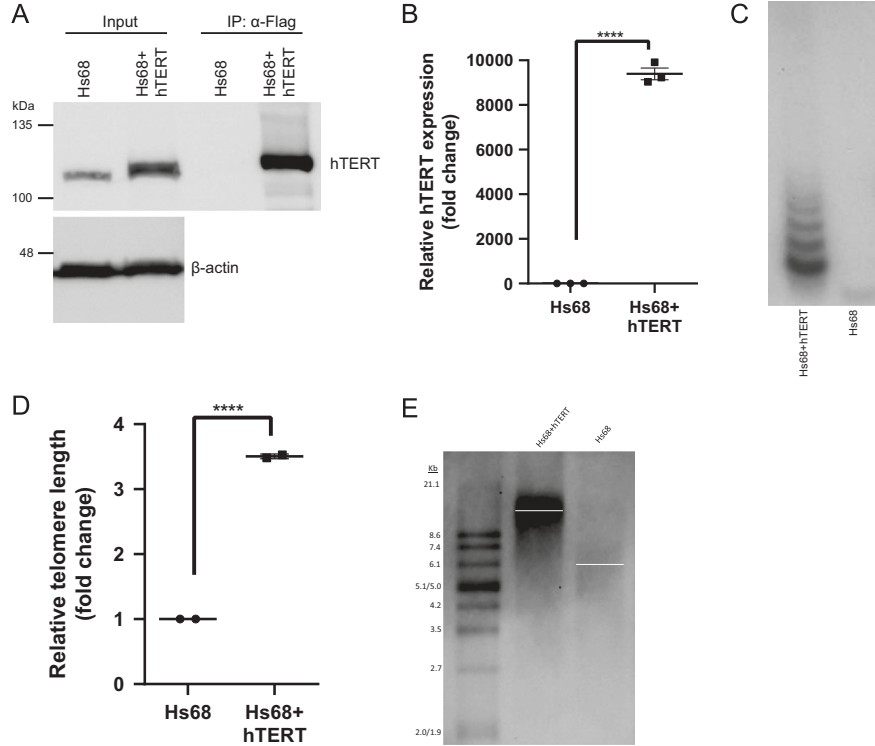

**Fig. 6 Confirmation of hTERT expression and telomere elongation in the generated cell line Hs68 + hTERT. a** Immunoprecipitation of Flag-hTERT in Hs68 and Hs68 + hTERT cells followed by western blot for hTERT protein. Low levels of hTERT were detected in input samples of Hs68, but no hTERT was detected after immunoprecipitation with Flag-antibody. In both input and immunoprecipitated samples of Hs68 + hTERT high levels of hTERT were detected, indicating successful infection with hTERT-plasmid. **b** RNA expression of hTERT in Hs68 and Hs68 + hTERT measured by qPCR. **c** Telomerase activity of hTERT in Hs68 and Hs68 + hTERT measured by Telomerase Repeated Amplification Protocol (TRAP). **d** Relative amount of telomeric DNA in Hs68 and Hs68 + hTERT measured by telo-qPCR. Upon hTERT overexpression, there is a 3.5-fold increase in average telomere length in Hs68 + hTERT compared to Hs68 cells. **e** Absolute telomere length in Hs68 (~5.3 kb) and Hs68 + hTERT (~18.9 kb) measured by Telomere Restriction Fragments (TRF) analysis. This matched the telo-qPCR data and a 3.5-fold increase in telomere length upon hTERT overexpression. Two-tailed unpaired *t*-tests were performed, *n* = 3 biologically independent experiments, ****P* < 0.0001. Source data used for the graphs in **b** and **d** can be found in Supplementary Data 2.

**Telomere count is lower in Hs68 + hTERT compared to Hs68.**
As noted before, the noise tolerance in ImageJ should be set to a value where approximately one point is counted per telomere in the control cell line when using 'Find Maxima' feature. However, independent from the noise tolerance used, a significant decrease in telomere count was observed in Hs68 + hTERT cells compared to Hs68 (Fig. 7f). As determined previously, a noise tolerance of 10 in Hs68 cells resulted in ~92 telomeres per cell. When applying this noise tolerance to Hs68 + hTERT, we observed an average of 79 telomeres per cell. Consistent with this relationship, when comparing the number of spots assigned in Imaris in 3D, Hs68 cells had an average of 73 telomeres per cell whereas Hs68 + hTERT had an average of 64 telomeres per cell (Fig. 7g). This suggests that there is likely to be an increased degree of telomere clustering in Hs68 + hTERT cells compared to parental Hs68 cells, potentially due to longer telomere length.

## Discussion
Telomere biology plays an important role in determining cell lifespan that is impacted by replicative senescence and immortalization. Wide-field microscopy is a standard method to visualize and measure individual telomere lengths in metaphase spreads when combined with 2D analysis techniques. While the telomere dynamics and characteristics of cells in metaphase have been examined to address many questions, much less is known about telomere structure in intact interphase cells. To facilitate such studies, we have outlined an accessible, rigorous and

reproducible strategy to investigate telomeres in greater detail within the nuclei of interphase cells in 3D. Due to the resolution limit of wide-field microscopes and the limitations of 2D maximum intensity projections of Z-stacks, performing standard segmentation methods on compressed 2D images was not suitable to accurately measure telomere area and intensity. While this might not be a problem for metaphase spread analysis, due to the 3D morphology of nuclei, the maximum intensity projection of a 3D stack may lead to two telomeres appearing as one and inaccurate telomere measurements.

A combination of 2D techniques in ImageJ and 3D analysis in Imaris were used to measure telomere number, length and volume in interphase cells using two biological samples, Hs68 and Hs68 + hTERT. We confirmed that Intensity Sum in 3D is an accurate and robust read-out for telomere length. Our telo-qPCR and TRF data showed a ~3.5-fold increase in telomeric DNA, which was consistent with Intensity Sum measurements. Assuming uniform hybridization efficiency, Intensity Sum values represent the summation of the intensity of each pixel within a given elliptical shape. In Imaris, the Intensity Max value is used to grow the spot and is placed in the centre of the spot. Therefore, Intensity Max is the same value as Intensity Centre and can be used to determine the brightest telomeres. When calculating the coefficient of variation (CV), a larger variation was observed in Hs68 + hTERT for Intensity Sum, Mean and Centre (Supplementary Table 2). This is not surprising since the telomeres in Hs68 + hTERT appear to be variable in both length and in size (Fig. 7e). When taking a closer look at the graph in Fig. 7e, the

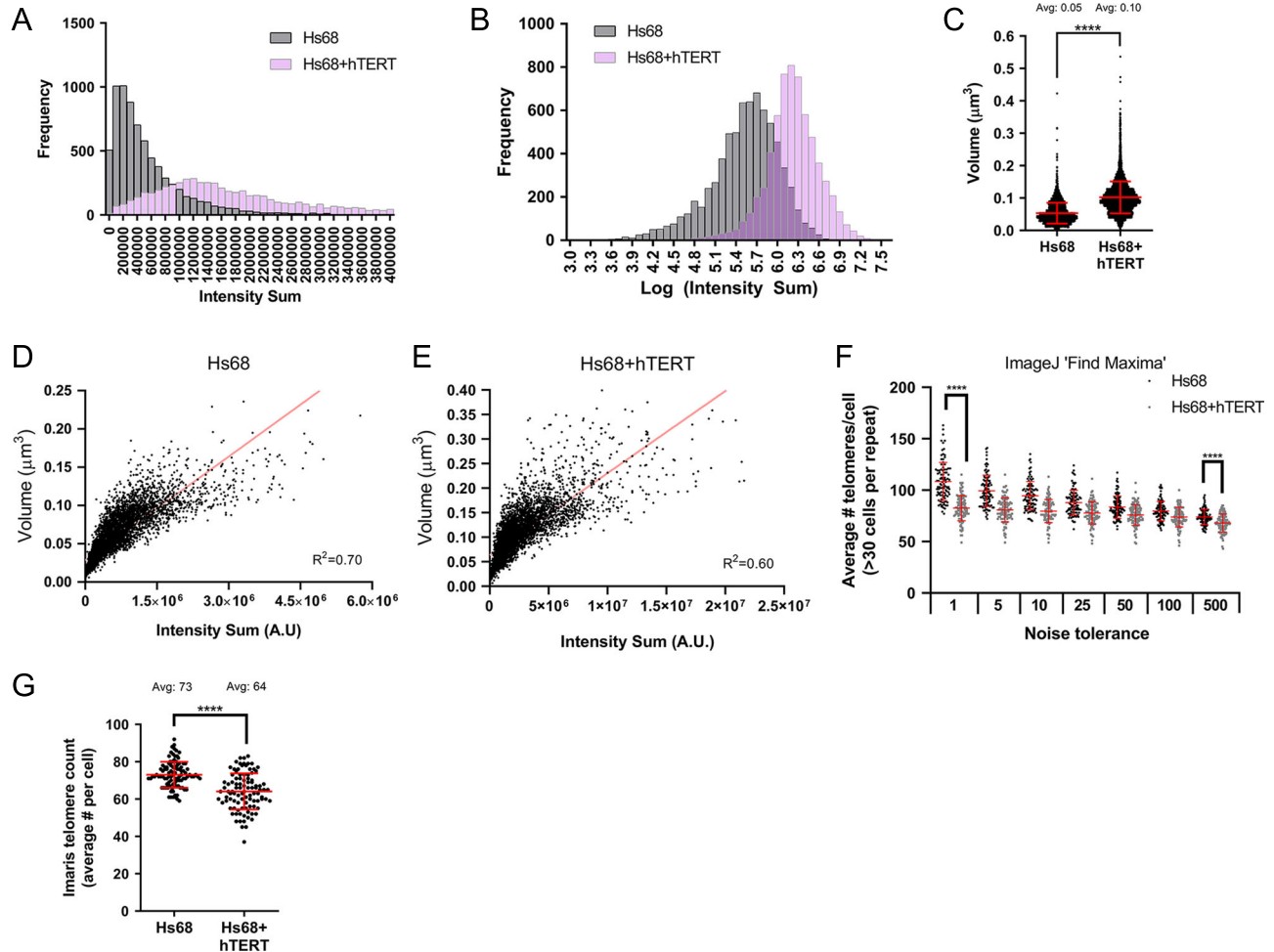

**Fig. 7** Hs68 + hTERT cells have longer and bigger telomeres when compared to Hs68. **a** Frequency distribution of the Intensity Sum of the individual spots assigned in Imaris in 3D. Intensity Sum represents telomere length of the telomeres in Hs68 and Hs68 + hTERT. **b** Histogram of the logarithm of the Intensity Sum of the spots. **c** Volume measurements of the individual spots in 3D. Telomeres in Hs68 + hTERT cells are on average twofold bigger when compared to telomeres in Hs68 cells. **d**, **e** The coefficient of determination, $R^2$, demonstrates a non-linear correlation between intensity and volume of individual spots in Hs68 and Hs68 + hTERT. A weaker $R^2$ is observed in Hs68 + hTERT cells compared to Hs68. Note the difference in $X$ and $Y$ axes when comparing Hs68 (**d**) with Hs68 + hTERT (**e**). **f** Telomere count in 2D using the Find Maxima setting in ImageJ and various noise tolerances. **g** Telomere count in 3D using Imaris 'Spots' feature. Hs68 + hTERT cells have on average less telomeres per cell compared to Hs68. Two-tailed unpaired $t$-tests were performed, $n = 3$ biologically independent experiments, ****$P < 0.0001$. Source data used for the graphs can be found in Supplementary Data 3.

telomere volume appears to be variable even while maintaining the same intensity value suggesting that telomeres of the same length can be more or less compact. This is in line with studies using super-resolution techniques and gold nanoparticle-fluorescent probes which have shown that telomeres of a certain length can vary in their diameter up to a fivefold range[24–27]. Due to the resolution limit of wide-field microscopes, the smallest resolvable telomeres after deconvolution have an elliptical volume of ~0.015 µm³ (Supplementary Fig. 2, Supplementary Table 3). In Hs68 cells we see that ~7% of telomeres appear smaller than 0.015 µm³ (Supplementary Fig. 4, Supplementary Data 4). However, the average volume of telomeres in Hs68 cells appears to be around 0.05 µm³ (Fig. 7c), which is a factor of four larger than the smallest resolvable telomere indicating this method provides relative volume measurements of telomeres in normal fibroblasts from the acquired images.

We observed a decrease in telomere count in both Hs68 and Hs68 + hTERT cells after 3D analysis of telomeres (Fig. 7g). When taking a closer look at the volumes of the individual telomeres, a bimodal distribution was observed in both Hs68 and

Hs68 + hTERT (Supplementary Fig. 4). This indicates that after a certain size, 0.07 µm³ in Hs68 cells, the analysed spots potentially no longer represent single telomeres. Since 92 telomeres were expected, yet on average only 73 telomeres per Hs68 cell were identified, we can conclude that 20–25% of the assigned spots represent >1 telomere (Supplementary Table 4). Similar observations have been made by other studies and suggested that telomeres have a tendency to associate, cluster or aggregate[21,49,50]. Largely, this can be interpreted as telomeres being too close together to be measured independently leading to a reduction in overall telomere count. Moreover, live-cell imaging studies have revealed the potential of telomeres to physically associate and dissociate from one another throughout the cell cycle of U2Os, HeLa and human primary cells[51,52], indicating that telomere clustering is a dynamic process. To our surprise, we observed a bigger decrease in telomere count in Hs68 + hTERT cells compared to parental Hs68 fibroblasts (Fig. 7f, g). Hs68 + hTERT-cells have ~3.5 times more telomeric sequence per cell than Hs68, which indicates that there may be spatial overlap of longer telomeres within the nucleus. Alternatively, this observation more

likely indicates that longer telomeres in Hs68 + hTERT cells have a greater propensity to cluster. Indeed, even though Hs68 + hTERT cells have ~3.5 times more telomeric sequence per cell than Hs68, we observed only a twofold increase in average telomere volume when comparing Hs68 with Hs68 + hTERT, which was independent of an increase in nuclear volume (Fig. 7c, Supplementary Table 5, Supplementary Data 4). With the development of spectral karyotyping (SKY) FISH, studies have shown that chromosomes tend to occupy defined regions within the nucleus which is referred to as chromosomal territories[53–56]. When chromosomes are forming territories in the nucleus, it is possible that telomeres that are near each other cluster together. Even though telomeres only make up ~1/4000th to 1/6000th of the total genome (based on 10–15 kb telomeres at the two ends of each of the 46 chromosomes in a $6 \times 10^9$ bp human genome), their distribution does not seem to be random. Given that telomeres are protected by high concentrations of the shelterin group of proteins[2] and contain TERRA RNAs[57], we speculate that longer telomeres show increased tendencies to cluster due to a liquid–liquid phase transition. This is also believed to contribute to the formation of numerous other subnuclear structures based upon concentration of particular proteins and RNA such as occurs in nucleoli and other membrane-less organelles.

In conclusion, here we presented a detailed workflow of how to image, process and analyse telomeres in 3D in interphase cells using a wide-field imaging system. We have applied this technique to two biological samples, diploid Hs68 and Hs68 + hTERT fibroblasts, to analyse telomere count, telomere intensity, and telomere size. This approach reveals that there appears to be both preferential compaction and/or clustering of longer telomeres. Although the exact mechanism responsible for clustering is unclear, our data and the high concentrations of shelterin proteins and TERRA RNAs found at telomeres suggest that clustering may be due to formation of domains resembling recently described membrane-less organelles in which liquid–liquid phase transitions promote formation of discrete domains[58].

## Methods

**Cell culture and generation of Hs68 + hTERT cell line**. All cell lines were directly obtained from ATCC and tested for mycoplasma regularly before performing the experiments. Primary human foreskin fibroblasts Hs68 (CRL-1635, ATCC) cells were grown in DMEM, 1 g/L glucose (Gibco, 11885–084) containing 10% fetal bovine serum (FBS). For stable Hs68 infection with hTERT-construct, detailed procedure was described elsewhere[59]. Briefly, HEK 293 cells (CRL-1573, ATCC) functioned as the packaging line using pBABE$_{puro}$-FLAG hTERT construct (1 μg), pCL-10A1 helper plasmid (1 μg), and X-tremeGENE 9 DNA Transfection Reagent (Roche) (3:1 ratio) to perform a double transfection. After each transfection, the viral supernatant produced by HEK 293 cells was filtered though an Acrodisc® syringe filters, HT Tuffryn® membrane, pore size 0.45 μm (Life Sciences), substituted with 10 μg/mL Polybrene/Hexadimethine bromide (Cat No. 107689; Sigma-Aldrich) and used for a double infection of Hs68 cells (passage 25). After the second infection, hTERT-infected cells were selected for using puromycin (0.5 μg/mL). To confirm telomerase protein expression, cells were harvested and used for hTERT immunoblotting, qPCR and Telomerase Repeated Amplification Protocol (TRAP).

**FISH staining**. To visualize telomeres, a PNA probe conjugated to a fluorophore was used to stain the telomeric DNA using FISH. The fluorophore for FISH (Alexa Fluor-647) had absorption and emission spectra that matched the filters available on the microscope. Other considerations are photostability, brightness and compatibility with other probes if multi-colour fluorescent experiments are conducted[60]. Since the PNA probe is highly specific to its complementary sequence and forms stable DNA–PNA hybrids, little to no off-target staining was observed. The staining was performed according to the protocol described in Lansdorp et al.[18] with some adjustments. Briefly, cells were seeded on coverslips and fixed using 3% paraformaldehyde (PFA)/2% sucrose for 10 min and permeabilized with 0.5% Triton X-100 for 4 min. Next, cells were pre-blocked for 10 min with 2% bovine serum albumin (BSA) and blocked for 30 min with 10% normal goat serum to reduce non-specific labelling of the probe to the nucleolus. After washing 3 × 5 min with phosphate-buffered saline (PBS) solution, the coverslips were dehydrated in a series of ethanol concentrations (70–80–90–100%) for 3 min and air-dried before incubation with 100 nM TelC647 conjugated PNA-probe (PNA Bio, F1013). The samples were denatured at 85 °C for 8 min before incubation for 2 h at RT in a humid, dark environment. Coverslips were then washed twice for 15 min at RT with 70% formamide/10 mM Tris with vigorous shaking and three times for 5 min at RT with 0.1 M Tris/0.15 M NaCl pH 7.5 containing 0.05% Tween-20 with shaking. DNA was stained with DAPI (Millipore Sigma, 1:10.000) in PBS for 8 min at RT, and the slips were cured overnight with ProLong Diamond Antifade Mounting solution (Thermo Fisher Scientific), which is designed to not shrink when cured and to increase axial resolution. In order to accurately compare different samples, the samples were prepared on the same day and followed the same incubation conditions and PNA concentration for each experimental run.

**Wide-field microscopy imaging**. A fully motorized Nikon Ti Eclipse inverted epifluorescence microscope equipped with Spectra X fluorescent light source with six solid state LED sources operating independently was used for imaging. The highest lateral resolution with a wide-field microscope is on the order of 200 nm, but the exact value depends on the numerical aperture (NA) of the objective as well as the wavelength of the emission fluorescence[42]. Therefore, it is important to work with objectives that have the highest possible NA (≥1.3). During our imaging, a ×60 oil immersion 1.4 NA objective was used and images were captured with a Hamamatsu Orca flash 4.0 v2 sCMOS 16-bit camera resulting in a pixel size of 0.108 μm/pixel. The system is controlled by Nikon NIS-Element Imaging Software (Version 5.00). Telomeres (PNA TelC647) were imaged using the 633 nm LED at low light intensity and longer exposure time (25%, 1 s) to avoid bleaching and photo-toxicity[61]. Z-stacks were acquired with 100 nm step size and 41 steps in total. For the dark frame images, the shutter was closed, and 15 images were acquired. For the flat-field correction, 1-mm-thick auto-fluorescent plastic slide (Cy5) were used (Chroma, Part No: 92001). For both dark frame and flat-field images, the same LED settings as for telomeres were used. Samples from the same experiment were imaged on the same day to compensate for day-to-day drift of the microscope. Samples from different conditions were interspersed in the imaging session as well.

**Immunoprecipitation, western blotting, qPCR**. Hs68 and Hs68 + hTERT cell lysates were used for FLAG-hTERT immunoprecipitation as described elsewhere[59]. Briefly, NP-40 lysis buffer (10 mM Tris-HCl pH 7.5, 1% (v/v) NP-40, 10% (v/v) glycerol, 1 mM EGTA, 1 mM MgCl$_2$, 150 mM NaCl) containing Protease Inhibitor Cocktail tablet (Roche) was used to lyse the cells and anti-FLAG M2 affinity resin (Sigma-Aldrich, Canada) was used to immunoprecipitate FLAG-hTERT. After incubation, proteins were eluted from the beads using SDS buffer. Flag-hTERT immunoprecipitates from Hs68 cells were detected via western blotting using rabbit monoclonal (Y182) to telomerase reverse transcriptase (1:1000; ab32020; Abcam) as previously described[62]. For qPCR, RNA was isolated using TRizol Reagent (Invitrogen) using the manufacturer's protocol. After incubation with chloroform and isopropanol, the precipitated RNA was washed with 70% ethanol and air dried before dissolving in RNAse-free ddH2O. cDNA was made using 2 μg RNA using qScript cDNA SuperMix reaction protocol (Quantabio). Specific primers for hTERT and GAPDH (house keeping gene) were used to determine the relative hTERT levels in Hs68 and Hs68 + hTERT cells.

**Deconvolution of 3D imaging data**. All images were deconvolved with Huygens Essential version 18.10 (Scientific Volume Imaging, The Netherlands, http://svi.nl), using the Classic Maximum Likelihood Estimation (CMLE) algorithm, with SNR: 40 and 50 iterations. For general parameters, the sampling intervals used were 108 nm for X and Y, and 100 nm for Z. For the optical parameters, 1.40 for numerical aperture and the refractive index of lens immersion oil was 1.515. The images were taken using a wide-field microscope with an emission wavelength of 647 nm. In the deconvolution wizard, the point spread function was automatically generated from the Z-stack. The background of the measured image was automatically determined, and the original image was corrected for both bleaching and unstable illumination.

**Statistics and reproducibility**. All statistical analyses were performed in Graph-Pad Prism 8. Measurements were taken from distinct samples and three independent, biological replicates were measured for Hs68 and Hs68 + hTERT ($n = 3$) with >30 cells per experimental condition. Two-tailed unpaired $t$-tests were performed and statistical significance is indicated in the figure legends.

**Reporting summary**. Further information on research design is available in the Nature Research Reporting Summary linked to this article.

## Data availability statement

The data that support the findings of this study are available from the corresponding author upon request.

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

## Acknowledgements

We acknowledge the resources provided by the Live Cell Imaging Laboratory. The Nikon Ti Eclipse inverted epi-fluorescence microscope system was purchased with funds from the International Microbiome Centre, which is supported by the Cumming School of Medicine at University of Calgary, Western Economic Diversification (WED) and Alberta Economic Development and Trade (AEDT), Canada. We thank Dr. Nicolas Ting for useful scientific discussions and Joel Glover for assistance with image processing and data analysis. This research was supported by funding from the Arnie Charbonneau Cancer Institute, Robson DNA Sciences Centre, and operating grants from the Canadian Institutes for Health Research (CIHR) to K.R. and the National Sciences and Engineering Research Council (NSERC) to T.L.B.

## Author contributions

N.A. performed the majority of experiments, analysed the images and wrote the manuscript. E.D. helped with statistical analysis and interpretation of the results. S.B. did TRAP assays, R.M.W. and P.C. assisted in study design and editing, and T.B. and K.R. directed the study, obtained financial support, assisted in study design and manuscript writing.

## Competing interests

The authors declare no competing interests.
