## [Peer Review File · Communications Biology]

Reviewers' comments:

Reviewer #1 (Remarks to the Author):

In this paper, the authors provide a detailed and very useful workflow on how to analyze telomeres by FISH using wide-field imaging systems. I am listing below specific points that should be considered.

1. Page 6, bottom. Perhaps you could also mention here as a test for quality that both sister telomeres in metaphase chromosomes should give the same staining intensity. They have basically the same telomere lengths.
2. Page 8: You mention that you adjusted the noise tolerance to achieve on average 92 telomeres per cell. However, I suspect that a subset of cells is in G2 of the cell cycle with twice as many telomeres. Why are G2 cells not seen or not considered?
3. Figure 6A: Hs68 should not express hTERT. What is the band in lane 1 in the Western blot? I presume it is an unspecific band but this should be specified.
4. Figure 6C: Telomere lengths measurements on Southern blots would be much more informative and should be added. This would provide information on absolute telomere lengths. Such data seem important for this methods paper in order to facilitate implementation of the technology by other researchers. In our experience, the qPCR methods are also not as quantitative as Southern blots. Alternatively, measurements by STELA could be provided.
5. For the presumed clustering of telomeres in Hs68+hTERT cells it would be reassuring to also analyze the telomeres in metaphase spreads as done in Figure 2, to exclude chromosome loss events or telomere end fusions.
6. Page 10: Application of FISH for large scale studies. I would cite here papers from laboratories that are known for good quality research (e.g. Lansdorp).
7. Figure legend of Figure 2: Spelling Metaphase

Reviewer #2 (Remarks to the Author):

This manuscript presents a new work-flow for telomere analysis in interphase cells using wide-field 3D microscopy. The authors show that, in one hTERT-immortalized line, there are a smaller number of telomere clusters than in the non-immortalized parental cells. These results have biological novelty because they suggest that longer telomeres have a higher propensity to cluster. This microscopy method thus promises to add complementary information to the other quantitative methods of telomere analysis that have been described, and will be a useful and welcome addition to the telomere field.

The methods are thoroughly described and support the conclusions. There are 3 minor criticisms that can be addressed experimentally, or via further discussion of the limitations of not performing these additional tests:

- 1) It was unclear if the background staining of the PNA probe was measured (e.g. after Bal31 or DNase digestion). If the authors have this data, they should include it in supplementary material.
- 2) The experiments include one cell line, which begs the question if the discrepancy in telomere volume and number is unique to this cell line. Have other matched primary and TERT lines been tested? If not, the authors can discuss the potential limitations of analyzing one parental line.
- 3) The authors state that they have quantified telomere length, however qPCR is non-linear under

certain conditions and it only measures relative telomere length. For absolute length, qFISH should be conducted in order to compare to the values obtained by their method. At the least, the authors could reiterate the limitations of qPCR and, as they correctly state in the figures, always include the qualifier "relative" when referring to "length".

Other minor comments:

-when stating the number of replicates of experiments, please specify if they are biological or technical replicates

-tailed t-tests are appropriate for comparisons between two samples. For experiments in which multiple samples are included, please use ANOVA. If no multiple group comparisons were made, please state so in the Methods.

Reviewer #3 (Remarks to the Author):

This manuscript by Adam et al. discusses mammalian telomere clustering and introduces the reader to protocol, imaging and image analysis of human telomeres in 2D and 3D modes.

General comments:

- 1) the concept of mammalian telomere clustering is not new.
- 2) Programs to analyze telomeres were described and published previously.

Specific comments.

1) Abstract and Title: the final conclusion of the abstract refers to the observations the authors made with Hs68 and hTERT-immortalized Hs68 cells. This should be included in the concluding sentence must reflect this. The generalized conclusion that "long telomeres" have a higher propensity to form clusters has not been conclusively shown with the use of a single cell line. Therefore, the conclusion should read...shows that longer telomeres in hTERT-immortalized Hs60 cells have a higher propensity to cluster,...". In this regard, the title is also an overstatement and should refer more specifically to the data the authors obtained when comparing Hs68 and hTERT-immortalized Hs68 cells.

2) Introduction.

a. The six shelterin proteins capping telomeres refer to human telomeres. In mouse, there are 7 shelterin proteins. Please edit to make this clear in your text.

b. The use of the term 'strain' is unusual for mammalian cell lines. It is more common in yeast or bacteria. This reviewer suggests using the term 'cell line' instead of strain.

c. No mention of existing telomere measurement tools is made in the Introduction. The question of novelty arises.

3) Methods. Some descriptions remain unclear and require clarifications;

a. The matching of fluorophores and filters "as closely as possible" is not an option but a must. "As closely as possible" is too vague. Each dye/fluorophore must be matched with an appropriate filter.

b. Image acquisition. This section is not clear on the x,y,z step sizes used during the 3D imaging process.

c. The use of a plastic slide (Chroma) to acquire a flat-field image is not optimal as the plastic slide does not accurately reflect the aberration from a glass microscopy slide (in addition, it does not have a cover slip).

d. Analysis of telomeres in 3D. Imaris (not a free program) is used to measure 3D telomere parameters. This is the first time the authors acknowledge the existence of other programs that

measure telomeres, but the list is not all inclusive. In addition, no comparison is made what is different/identical with the Imaris-written program and the other existing ones. The authors simply write 'similar'. This is not sufficient for this reviewer to judge the differences/similarities. In fact, if the program is similar, why do we need it? What is the innovation? Moreover, to appropriately judge a program, it is recommended running it side by side with an existing published (peer-reviewed) program.

e. Sample preparation. There are protocols for the 3D preparation of cells. These were established by the Cremer lab and they do not allow for steps that involve air drying of samples (which flattens the nucleus). In addition, Prolong Gold, which the authors use as antifade, is not recommended as it artificially flattens the nuclei. It is highly recommended to work with Glycerol-based mounting media in order to preserve the 3D structure of the nucleus (such as Vectashield).

f. Exposure times used for the telomere signals. This reviewer is surprised about the long exposure times required for 3D imaging. One second exposures are very high. Millisecond exposures are common in the literature. Since the authors use primary Hs68 cells and Hs68 cells with hTERT (longer telomeres), the exposure time is worrisome.

4) Discussion.

a. The authors acknowledge that telomere clustering has been observed before.

b. The authors do not cite earlier work on chromosome territories but pick one citation. More are needed to reflect the contribution of pioneers in the field, such as Cremer and colleagues.

c. hTERT-immortalized cells are called diploid, but we have not seen any SKY karyotype information. hTERT-immortalized cells, especially 40 passages or later into their generation may have acquired chromosomal rearrangements. SKY karyotypes should be included for both Hs68 and hTERT-immortalized Hs68 cells.

5) Additional comments.

a. Figure 3, Figure Legend. "E) Deconvoluted images..." – it should say "deconvolved images".

b. Figure 4. Several details are missing; TelC647 has not been defined; Panel C shows four images, which have not been described in detail.

c. Figure 5. What are the smeared lines in panel D?

We thank the reviewers for taking the time they have taken to do thorough reviews of our manuscript and for pointing out deficiencies and areas for improvement in the original manuscript. In the sections below we have listed all of the reviewers points, and have responded to each in *italics*. We have also included a version of our manuscript with all changed highlighted in red type that help identify the individual responses and changes made.

Reviewers' comments:

Reviewer #1 (Remarks to the Author): In this paper, the authors provide a detailed and very useful workflow on how to analyze telomeres by FISH using wide-field imaging systems. I am listing below specific points that should be considered.

1. Page 6, bottom. Perhaps you could also mention here as a test for quality that both sister telomeres in metaphase chromosomes should give the same staining intensity. They have basically the same telomere lengths.

This has been done and is now noted in the test with addition of reference 36.

2. Page 8: You mention that you adjusted the noise tolerance to achieve on average 92 telomeres per cell. However, I suspect that a subset of cells is in G2 of the cell cycle with twice as many telomeres. Why are G2 cells not seen or not considered?

We have addressed this point on page 8 by the addition of: "Since we were working with Hs68 cells that are known to have a diploid genome, we adjusted the noise tolerance to achieve on average 92 telomeres per cell in G1 phase. Cells in S and G2, measured by their DNA content, were excluded from the analysis."

3. Figure 6A: Hs68 should not express hTERT. What is the band in lane 1 in the Western blot? I presume it is an unspecific band but this should be specified.

This is a good point and one that we also struggled with but it appears that despite not displaying easily detectable activity, there are low levels of hTERT expressed in normal diploid fibroblasts as noted in the paper from the Hahn lab (Masutomi, K et al. Cell, Volume 114: 241-53, 2003) now included in the references (<https://www.ncbi.nlm.nih.gov/pubmed/12887925>).

4. Figure 6C: Telomere lengths measurements on Southern blots would be much more informative and should be added. This would provide information on absolute telomere lengths. Such data seem important for this methods paper in order to facilitate implementation of the technology by other researchers. In our experience, the qPCR methods are also not as quantitative as Southern blots. Alternatively, measurements by STELA could be provided.

We agree that different methods have their strengths and limitations and so we have added the data requested in our revised Figures 6D and 6E.

5. For the presumed clustering of telomeres in Hs68+hTERT cells it would be reassuring to also analyze the telomeres in metaphase spreads as done in Figure 2, to exclude chromosome loss events or telomere end fusions.

This is now done and shown in the revised and expanded Figure 2.

6. Page 10: Application of FISH for large scale studies. I would cite here papers from laboratories that are known for good quality research (e.g. Lansdorp).

This has been done with two references added.

7. Figure legend of Figure 2: Spelling Metaphase

We thank the reviewer for catching this and it has been corrected.

Reviewer #2 (Remarks to the Author):

This manuscript presents a new work-flow for telomere analysis in interphase cells using wide-field 3D microscopy. The authors show that, in one hTERT-immortalized line, there are a smaller number of telomere clusters than in the non-immortalized parental cells. These results have biological novelty because they suggest that longer telomeres have a higher propensity to cluster. This microscopy method thus promises to add complementary information to the other quantitative methods of telomere analysis that have been described, and will be a useful and welcome addition to the telomere field.

The methods are thoroughly described and support the conclusions. There are 3 minor criticisms that can be addressed experimentally, or via further discussion of the limitations of not performing these additional tests:

1) It was unclear if the background staining of the PNA probe was measured (e.g. after Bal31 or DNase digestion). If the authors have this data, they should include it in supplementary material. *We believe that the metaphase spreads shown in the new Figure 2 show how specific the PNA staining is, and the intensity of the signals was also thresholded to help address this point. In addition, the correct number of telomeres were detected in metaphase spreads from primary cells so it does indeed appear that the signal is specific and background staining does not represent a confounding variable. This is noted on pages 6 and 7 of the revised manuscript.*

2) The experiments include one cell line, which begs the question if the discrepancy in telomere volume and number is unique to this cell line. Have other matched primary and TERT lines been tested? If not, the authors can discuss the potential limitations of analyzing one parental line. *The reviewer is correct that we have done these experiments in one cell strain and an immortalized/telomerized version of the same strain. Although these are genetically stable, it is true that we cannot generalize to other cell strains/lines at this time. Our study is a proof-of-principle and subsequent strains and/or lines of cells will be followed up on in subsequent studies.*

3) The authors state that they have quantified telomere length, however qPCR is non-linear under certain conditions and it only measures relative telomere length. For absolute length, qFISH should be conducted in order to compare to the values obtained by their method. At the least, the authors could reiterate the limitations of qPCR and, as they correctly state in the figures, always include the qualifier "relative" when referring to "length". *The reviewer is correct about the qPCR method being a relative measure and so we have provided an additional set of experiments to look at telomere length using an unrelated method (TRF assay). These data that we have included agree quite well with results from qPCR and so we have validated qPCR values of difference and have tied those numbers to more accurate estimations of telomere length.*

Other minor comments:

-when stating the number of replicates of experiments, please specify if they are biological or technical replicates

We thank the reviewer for the reminder and now include those data in figure legends.

-tailed t-tests are appropriate for comparisons between two samples. For experiments in which multiple samples are included, please use ANOVA. If no multiple group comparisons were made, please state so in the Methods.

We have added the additional, and important details of the statistical analyses requested in the figure legends.

Reviewer #3 (Remarks to the Author):

This manuscript by Adam et al. discusses mammalian telomere clustering and introduces the reader to protocol, imaging and image analysis of human telomeres in 2D and 3D modes.

General comments:

1) the concept of mammalian telomere clustering is not new.

We agree, but the idea of having longer telomeres cluster to a greater degree is new to our knowledge. We have pointed this out in the introduction and discussion of the revised manuscript.

2) Programs to analyze telomeres were described and published previously.

This is indeed the case and we are aware of these programs. What we present is a simple-to-use workflow that we have not seen published to date and point out some biologically interesting observations using it.

Specific comments.

1) Abstract and Title: the final conclusion of the abstract refers to the observations the authors made with Hs68 and hTERT-immortalized Hs68 cells. This should be included in the concluding sentence must reflect this. The generalized conclusion that “long telomeres” have a higher propensity to form clusters has not been conclusively shown with the use of a single cell line. Therefore, the conclusion should read...shows that longer telomeres in hTERT-immortalized Hs60 cells have a higher propensity to cluster,...”. In this regard, the title is also an overstatement and should refer more specifically to the data the authors obtained when comparing Hs68 and hTERT-immortalized Hs68 cells.

Although we believe that our observation regarding telomere length and clustering will hold in other cell strains, the reviewer is correct and so we have altered the title to reflect this.

2) Introduction.

a. The six shelterin proteins capping telomeres refer to human telomeres. In mouse, there are 7 shelterin proteins. Please edit to make this clear in your text.

We thank the reviewer for reminding us of this and have added the additional information.

b. The use of the term ‘strain’ is unusual for mammalian cell lines. It is more common in yeast or bacteria. This reviewer suggests using the term ‘cell line’ instead of strain.

The term strain in mammalian cells refers to a primary, non-immortalized cell types as compared to immortalized or cancer-derived cell lines. Since this differentiates these types of cells we prefer to maintain this nomenclature.

c. No mention of existing telomere measurement tools is made in the Introduction. The question of novelty arises.

We agree although other existing telomere measurement tools are mentioned in the results section. In this paper we provide a workflow that can be used with several of the software packages for measuring telomere length, and show how the workflow can be used to make novel observations.

3) Methods. Some descriptions remain unclear and require clarifications;
a. The matching of fluorophores and filters “as closely as possible” is not an option but a must. “As closely as possible” is too vague. Each dye/fluorophore must be matched with an appropriate filter.

We agree and the language has been changed to emphasize this point.

b. Image acquisition. This section is not clear on the x,y,z step sizes used during the 3D imaging process.

Again, we agree and the suggested clarifications have been added to clarify this point.

c. The use of a plastic slide (Chroma) to acquire a flat-field image is not optimal as the plastic slide does not accurately reflect the aberration from a glass microscopy slide (in addition, it does not have a cover slip).

This is indeed true. Using dye solutions is a better and more accurate way to adjust for shading correction. We have changed our practice, especially for green (fluorescein) and red (rhodamine) channels. We have added a reference in the text to these dyes. In our experiments, Alexa647 was used and there are no cheap dyes available to correct in the Cy5 channel (see link below). Therefore, we chose to use the Chroma autofluorescent plastic slides. Chroma autofluorescent plastic slides are intended to determine the consistency and evenness of illumination of excitation light on a fluorescent sample. For the purpose of our experiment it worked well. Furthermore, cells around the edges of the FOV (~5% of the total cells) were avoided. These issues have been addressed in the text and we have added two additional references to document this.

<http://nic.ucsf.edu/blog/2014/01/shading-correction-of-fluorescence-images/>

d. Analysis of telomeres in 3D. Imaris (not a free program) is used to measure 3D telomere parameters. This is the first time the authors acknowledge the existence of other programs that measure telomeres, but the list is not all inclusive. In addition, no comparison is made what is different/identical with the Imaris-written program and the other existing ones. The authors simply write ‘similar’. This is not sufficient for this reviewer to judge the differences/similarities. In fact, if the program is similar, why do we need it? What is the innovation? Moreover, to appropriately judge a program, it is recommended running it side by side with an existing published (peer-reviewed) program.

We are aware of these programs and have referenced them in the text. In our manuscript we are not suggesting that we are developing a new software. We use software that is already available (e.g. Imaris and ImageJ in our case) and are simply providing a rigorous workflow/approach so that there is consistency in how people analyse telomeres in their images. We believe that this can be done with TeloView, or it can be done with any other software that an imaging facility has available. Furthermore, since TeloView is a licensed software and we don't have their software, we can't comment on the differences/similarities of that software and our approach in depth. We tried to contact TeloView to get more information about obtaining a license, but we received no response from 3DS. We understand that the TeloView software can identify and measure telomeric features such as intensity, count and localization within a cell. This can also be done with other software such as Imaris that we used,

but again our manuscript describes a rigorous approach using one of the different software packages available and does not attempt to replace features of any of them.

e. Sample preparation. There are protocols for the 3D preparation of cells. These were established by the Cremer lab and they do not allow for steps that involve air drying of samples (which flattens the nucleus). In addition, Prolong Gold, which the authors use as antifade, is not recommended as it artificially flattens the nuclei. It is highly recommended to work with Glycerol-based mounting media in order to preserve the 3D structure of the nucleus (such as Vectashield).

In our study we used Prolong Diamond, which is designed to not discolor or shrink when cured and stored long term. Furthermore, one of the key features of Prolong Diamond is that it increases the axial resolution in immunofluorescence when compared to Vectashield due to near-perfect optical path (1.47 RI). This mounting media has been used in other studies as well to analyse 3D structures as noted in the references below.

<https://www.thermofisher.com/order/catalog/product/P36965>
https://www.molbiolcell.org/doi/full/10.1091/mbc.E15-07-0461?url_ver=Z39.88-2003&rfr_id=ori%3Arid%3Acrossref.org&rfr_dat=cr_pub%3Dpubmed&

Regarding comparison to the Cremer paper mentioned regarding the 3D preparation of cell nuclei: (<https://www.ncbi.nlm.nih.gov/pubmed/18951171/>) we noted additional steps were added such as the use of liquid nitrogen and pepsinization. We did not use pepsinization in our FISH protocol and cells were fixed with PFA which we believe can stabilize and preserve 3D configuration and structure. The dehydration steps and air-drying happened after fixation and we therefore did not believe that we have issues that would invalidate our 3D imaging and analysis.

f. Exposure times used for the telomere signals. This reviewer is surprised about the long exposure times required for 3D imaging. One second exposures are very high. Millisecond exposures are common in the literature. Since the authors use primary Hs68 cells and Hs68 cells with hTERT (longer telomeres), the exposure time is worrisome.

*We agree in principle with the reviewer, but PNA-647 is a somewhat unstable probe so is prone to bleaching. We have elaborated on this in the methods section and include an additional reference (Mubaid, F., & Brown, C. (2017). Less is More: Longer Exposure Times with Low Light Intensity is Less Photo-Toxic. *Microscopy Today*, 25(6), 26-35).*

4) Discussion.

a. The authors acknowledge that telomere clustering has been observed before.

Indeed – we have elaborated on this in the discussion.

b. The authors do not cite earlier work on chromosome territories but pick one citation. More are needed to reflect the contribution of pioneers in the field, such as Cremer and colleagues.

We have added additional references as suggested by the reviewer.

c. hTERT-immortalized cells are called diploid, but we have not seen any SKY karyotype information. hTERT-immortalized cells, especially 40 passages or later into their generation may have acquired chromosomal rearrangements. SKY karyotypes should be included for both Hs68 and hTERT-immortalized Hs68 cells.

Several groups including Shay and Wright, Harley, Weinberg and Benchimal have looked at this in immortalized fibroblasts and found them to be karyotypically normal and we have added this information in the form of a metaphase spread in figure 2 that confirms prior observations.

5) Additional comments.

a. Figure 3, Figure Legend. “E) Deconvoluted images...” – it should say “deconvolved images”.
We thank the reviewer for pointing this out and have changed the wording accordingly.

b. Figure 4. Several details are missing; TelC647 has not been defined; Panel C shows four images, which have not been described in detail.

These changes have been added as suggested.

c. Figure 5. What are the smeared lines in panel D? Figure legend will be explained.

We have explained this as suggested in the revised manuscript.

We hope that our responses address the points made the reviewers sufficiently, and we thank the reviewers for their time and insights regarding our work.

REVIEWERS' COMMENTS:

Reviewer #1 (Remarks to the Author):

The authors have addressed all my comments very well. I recommend acceptance of the paper.